# Mechanical Evaluation of Titanium Plates for Osteoesynthesis High Neck Condylar Fracture of Mandible

**DOI:** 10.3390/ma13030592

**Published:** 2020-01-27

**Authors:** Rafał Zieliński, Marcin Kozakiewicz, Bartłomiej Konieczny, Michał Krasowski, Jakub Okulski

**Affiliations:** 1Department of Maxillofacial Surgery, Medical University of Lodz, 1stGen. J. Hallera Pl., 90-647 Lodz, Polandjakub.okulski@gmail.com (J.O.); 2Material Science Laboratory, Medical University of Lodz, 251st Pomorska, 92-213 Lodz, Poland; bartlomiej.konieczny@umed.lodz.pl (B.K.); michal.krasowski@gmail.com (M.K.)

**Keywords:** condylar fracture, titanium plates, high neck fractures

## Abstract

Background: In the literature no information about plates for the high-neck mandibular condylar osteosynthesis could be found despite that 30 plate designs have been published. The main course consider the basal condylar or diacapitular fractures. The aim of the study was to test mechanically all available designs (only 4 of 30 was proper) on polyurethane mandibles using an individually designed clamping system. Methods: Forces required for a 1 mm displacement of fixed fracture fragments and incidents of screw loosening were recorded. Results: It has occured that dedicated plates for fixation are much weaker than set of two straight plates (*p* < 0.0001). General observation is the bigger plate and more screws, the better rigid stable osteosynthesis of mandibular condyle, however, there are limitations in plates design for high-neck fractures resulted in restricted operation field. Conclusion: Double straight plates occured to be the best mechanical fixation for high-neck fractures of the mandibular condyle. Maybe other existing plates could be used but only after prebending or that fracture required novel dedicated plates design.

## 1. Introduction

### 1.1. Epidemiological Information

Managment of mandibular fractures, especially condylar, are still a debateble issue for maxillofacial surgeons [1,2,3]. Some researches have prooved that surgical management have better outcomes than non-invasive treatment. Masticatory function in patients who underwent treatment by means of plates and screws were better. Nevertheless, in surgically treated patients choosing of particular plates in open reduction and condylar osteosynthesis is not clear because there are no guidelines in the literature [2,3,4,5,6,7,8,9]. Mandibular fractures are a part of the great majority of facial traumas. Condylar fractures comprise 17.5–52% of mandibular fractures according to the published large series. Most of the fractures are condylar base fractures. [10,11,12,13,14] In the literature there is no study showing which type of osteosynthesis plate should be used for management of low-neck and high-neck mandibular condyle fractures. Some authors admit that application of 2 plates—can be very demanding and is not always possible [15].

### 1.2. Surgical Procedures

Described approaches to mandibular condylar region are mainly preauricular and retromandibular. Those methods of dissections in order to treat the condylar neck and base fractures are debated and some authors described the approaches very precisely. [16] Authors claim that the main reason of choosing appropriate plating system is not the mechanical endurance but only restricted visibility and hardships resulted from low accessibility of this region. [17] However, no studies could be found with comparison of osteosynthesis plates dedicated to condylar neck fractures.

### 1.3. Biomechanics of the Mandible

The mandible has the same mechanical construction as lever class III including the fulcrum of rotation the condyle. Masticatory muscles exert forces and the load is transferred to teeth. [18,19] Muscles that play the main role in transmission of forces are following: masseter and medial pterygoid muscles together initate movements superiorly and anteriorly from the angle of the mandible. Temporalis muscle is attached from the coronoid process and directed superiorly and slightly posteriorly. Lateral pterygoid runs from the condyle anterior and medial direction. [20] Above muscles are not alone in playing the role in mandible movements but their role is the most significant in consideration of mandibular condylar fractures. Some other muscles also play the role in the movements of the mandible but they are negligible low. During jaw movements forces generated from masticatory muscles make compression and tension in the bone. When bone fracture appears, it means forces were much higher than physiologically. [21] The aim of diminish fractures is osseosynthesis that could resist to physiologic load and to endure tensions and compressions exerting on the condylar neck.

### 1.4. Osseosynthesis Plates

The prime aim of the study was to analyze in fatigues tests plates designed for high neck condylar fractures. Authors of this article proved that if an osseosynthesis plate is not available on the market it is not equal that a plate has worse mechanical endurance than those that are easy to buy. A lot of plates manufactured for mandibular condylar fractures fixations are available. In the literature it is impossible to find out which type of designs is proper for high-neck fractures.There is almost no data in the literature about comparison of osteosynthesis plating system. [4,5,6,7,8,9,10,11,12,13,14] Recently, 30 plates design have been compared in basal condylar fracture osseosynthesis. [22] Four plates’ designs among 30 have been concluded as those that could withstand screw pull-out and displacing force.

### 1.5. Aim

In high-neck condylar fractures only four plates might be used because of anatomy of condylar head. Aim of the article was to define appropriate plate design among available to high-neck fracture.

## 2. Materials and Methods

### 2.1. Mandibles

Mandible model made of solid foam used in the article (Figure 1a,b). Biomechanical testing outcomes depend on different density and elastic modulus of bone. [23,24] In the literature it is shown that polyurethan mandibles have been proved to be material of choice in orthopaedic implant testing, especially in fractures, and has been confirmed by the American Society for Testing and Materials. [25,26] The most natural would appear to be cadaver bone, but those differ from each other so results of biomechanical fatigue tests would not be standardized. [24] Solid polyurethane material has properties comparable to the human cancellous bone and it is widely used as an ideal medium to mimic human cancellous bone. In our study, polyurethane mandibles (Sawbones, Vashon, WA, USA: density 0.16 g/cc, compression modulus 58 MPa) were utilized as models for fatigued mechanical tests [27,28,29,30].

### 2.2. Plates

Among all available for rigid fixation of condylar process of mandible plates, only four designs can be applied for high-neck condylar fixation due to anatomical structure of head of condylar process. Some of the other plates [22] could be used, however, manual bending would be necesary and it would change physical properties the osteosynthesis. That is why only four plates were included into this study (Figure 2, Table 1). All plates were cut by laser from medical certified titanium sheets (grade 23, 1-millimeter thickness).

The condylar process was cut in level of typical head-neck condylar fracture in each model according to the newest classification. [31] Proximal head and distal ramus fracture segment were fixed by plate and the same 6-mm length self-tapping screws of 2.0 system. In each of 7 mandibles for plate design, drill 1.5 mm in width was used before filling plate holes by screws.

### 2.3. Simulation Set

Forces on temporomandibular joint were simulated according to the literature. [32] 15° inferiorly in the saggital and 10° laterally in the coronal plane mandibles were solid stabilized by screws on individual base plate (Figure 3). The plate was 1 mm trick made in stainless steel screwed on 0.7 m × 0.6 m tilted Block with 4 × M6 holes for stabilisation with screws. In such a construction forces were generated upwards, forwards, and medially.

All fatigues tests were used by means of Zwick Roell Z020 universal strength machine (Zwick-Roell, Ulm, Germany). Loading force was 1N and velocity of piston was 1 mm/min. All the compressive forces were pointed to the condyle. Special Instron software recorded relationship between load and displacement, load for permanent deformation, and maximum load at fracture.

Irreversible change of shape was described as the start point that the load-displacement relationship became non-linear. The moment when the highest load was recorded just before suddenly decrease was called maximum load (Figure 4).

According to the previous study [22] the Plate Design Factor was calculated:

Plate Design Factor = 0.850954 x Height (mm) + 0.846751 × Width (mm)+ 0.936732 × Plate (mm^2^) + 0.848039 × Total fixing screws.

### 2.4. Statistical Analysis

Height, width, plate surface area, plate design factor required force for one milimeter displacement in fracture line after plate fixation were recorded for interpretation of the experimental data.

Software used for statistics was Statgraphics Centurion 18 (Statgraphics Technologies Inc. The Plains, Virginia, USA). Kruskal–Wallis test was applied for between design comparisons. Independence Chi-Square tested categorical variables. Indicating the best plates were done on objective description.

## 3. Results

Only double plate fixation survived to the end of the experiment (Table 2). Three other osteosynthesisplates design lost screws in proximal fragment, as well as in distal fragment. As far as the plate integration was considered, the double plates broke during loading (in 160.6±27.3N) contrary to three other plates. Only two single plates tendP to break after rigid screws fixation. However, double plates fixation endured the highest loading (14.02±1.24 N/mm; *p* < 0.05) comparing to the three small plates (Figure 4). Plate Design Factor calculation that is valuable numerical description of plate design took into account the characteristics of the plate constructions as dimensions and number of screws possible to use (Table 3; Figure 5).

## 4. Discussion

### 4.1. Plates Combination

In the paper, double plates (plate design 20] has been proved as osseosynthesis material thanks to which excellent stability might be achieved. These plates have been well known [22,33] and derived from combination of biomechanical plates location and up to 4-screw fixation in the proximal fragment i.e. upper fragment [34]. However, till now it was impossible to find in the literature that scientifically prove double plate osteosynthesis is gold standard in high-neck fracture. By means of evidence performed in this study we proved that this thesis is true. Unfortunately, the main drawback of 2 plates number 20 used together is necessity of holding two separate plates firmly in very hard operating space. In our previous study we proved that total osseosynthesis plate surface is related with dedicated plate, and relation that the bigger plate is, the higher forces are required to 1-millimeter movement in fracture line (*p* < 0.05). The correlation coefficient equalled 0.58, indicating a moderately strong relationship between the plate surface area and the displacing force, points to the valuable feature of plates design which plays significant role in stable osteosynthesis (*p* < 0.05) [22].

Very important factor that we decided to include in our criteria was to choose plates dedicated anatomically for high neck fractures. It means we excluded plates that could be used but only after bending intrasurgically or before on patient-specific 3D stereolithographic condylar process. [35] Some additional experiments are needed to prove if other available plates could be strong enough to endure mastication forces in healing process. Without pre-bending many of presented plates design are not feasible for use because they are too big for high neck condylar process. Rigidity of plates is of paramount importance for every inventor, manufacturer, user or doctor for ORIF (open reduction and rigid internal fixation). Apart from the design, authors of the study wanted to describe physical properties of the condylar plates (grade of titanium alloy, annealing process, Young module, etc.), and asked manufacturers by phone and e-mails, however, any of the manufacturers did not answer our requests for information. Our study has not only been written directly doctors, surgeons but also by inventors, constructors, and medical designers. Authors want to emphasize the importance of design and present first worldwide comparison of all known designs of plate for high neck fractures. Obviously, it is still the open question: how in annealing the alloy, it may be better to use zirconium-molybdenum alloys or titanium-niobium alloys; is 0.9 mm thickness enough, or is 1.3 mm safer and/or more rigid? Those fascinating questions should answered the future. Moreover, comparison of new with smaller dimensions of plates with narrower screws with available plates but after bending could be interesting study, especially in high neck fractures where intrasurgical access is restricted. Some additional clinical papers are needed to compare the usage of 2 separate plates with others given that endocranial hardships such as restricted operation vision, time of screwing plates, import and anatomical structures that make screwing two plates harder to perform than one single.

### 4.2. Finite Element Method (FEM)

The rehabilitative dentistry has always paid particular attention to the detailed analysis application of occlusal forces, the distribution of tensional forces, and stress dissipation, asbiomechanical factors influence the prosthetic success substantially. Nowadays it is possible to design patient-specific plates for condylar fractures. [36] High technology in computing and manufacturing processes allowsurgeons to design a patient specific implant (PSI) to fit a particular fracture, including high neck, and to then before manufacturing analyze its performance in finite element analysis before screwing it in the patient. Thus a few different shapes can be quickly evaluated and the plate with the optimal outcome selected.

Because any dedicated plate was not confirmed biomechanically proper for osseosynthesis of the high neck fracture, it should be better fixation material that we’re looking for. In the literature it might be found that in high neck condyle fracture the most prone to fixation of the fracture was osseosynthesis by means of 3 screws. The area of stress in condylar head was negligibly low σ = 108 MPa and stress distribution in the rest of mandible was almost the same as in untouched bone in mastication process. Some authors have proved that one screw stabilization was the reason of relatively high displacement of bone ends, 558±245μm, whereas in 2 screws displacement was 218±81μm, and for 3 screws it was 217±144μm [34].

### 4.3. Plate Design Factor (PDF)

In our previous study where we compared plates systems in basal condylar fractures, probabilistic neural network PNN procedure pointed that plates of construction described by PDF over 300 were the most resistant to screw pull-out as well displacing force. However, in our present article PDF observed in tested fixations are relatively low comparing to results of basal condylar fracture fixations [22]. There was no result over 300, and only the double plate fixation exceeded 200. On one hand, it means that there is no proper dedicated plate for high neck fracture osteosynthesis, on the other hand, it is urgent matter to invent the effective one plate for this challenging fracture.

The suggested Plate Design Factor (PDF) should be indicator for forthcoming osseosythesis plates. It can be simply measured and compared with others under the circumstance 1-millimeter movement in fracture line will be considered. The discrimination power of that factor is such high as even very similar plates in this study can reach significant difference from another (and can be individually considered).

The experiment cannot prove advantage of the osseosynthesis plate number 11 which was the worst among all 4 plates (the lowest force required for 1-millimetere displacement, easiness in screw pull out, the lowest Plate Design Factor). For practitioners it is crucial to know whether plates must not apply or use only with consideration of reduction of forces in occlusion in high-neck fractures (i.e. plate number 11). It is possible that other osseosynthesis plates that authors of this article examined in another publication in lower condyle fracture could be as good as plater number 20 or even better. The question remains as to if the bending of other plates designed for lower condyle fractures play significant role in mechanical properties of titanium alloy grade 23? In this study any bending of titanium plates were performed in order not to diminish titanium properties but maybe individual bending of plates will not change significantly endurance of material. Comparison of all available plates after preparation such as changing the shape of plates in order to obtain the highest possible bone surface plate contact should be the next mechanical research valuable for treatment patients as well as to know mechanical properties of titanium alloy grade 23.

The next problem are study restrictions. Despite the fact the mechanical properties of the foam models were comparable to real mandible, some discrepancies in the structure of the materials were observed. For example artificial models have an almost homogeneous pore size, whereas in human mandibles consist of complex texture filled with different size of pores. It might play the role in compression efficacy and torque of the screws. The outcomes of our research were made on asingle-density foam polyurethane bone; but the biomechanical properties of the screws changes with the bone density environment. [29] Further fatigue tests and FEM analysies with changed shape of other plates are necessary.

Concluding, the double straight plates are the best mechanical fixation for high-neck fractures of the mandibular condyle. The limitation of its application significantly limited area for double 2.0 system plate fixation. Maybe other existing dedicated plates could be used, but only after prebending. That shape manipulation influences the mechanical properties of the material. That is the exploring field for novel dedicated plates designs searching.

## Figures and Tables

**Figure 1 materials-13-00592-f001:**
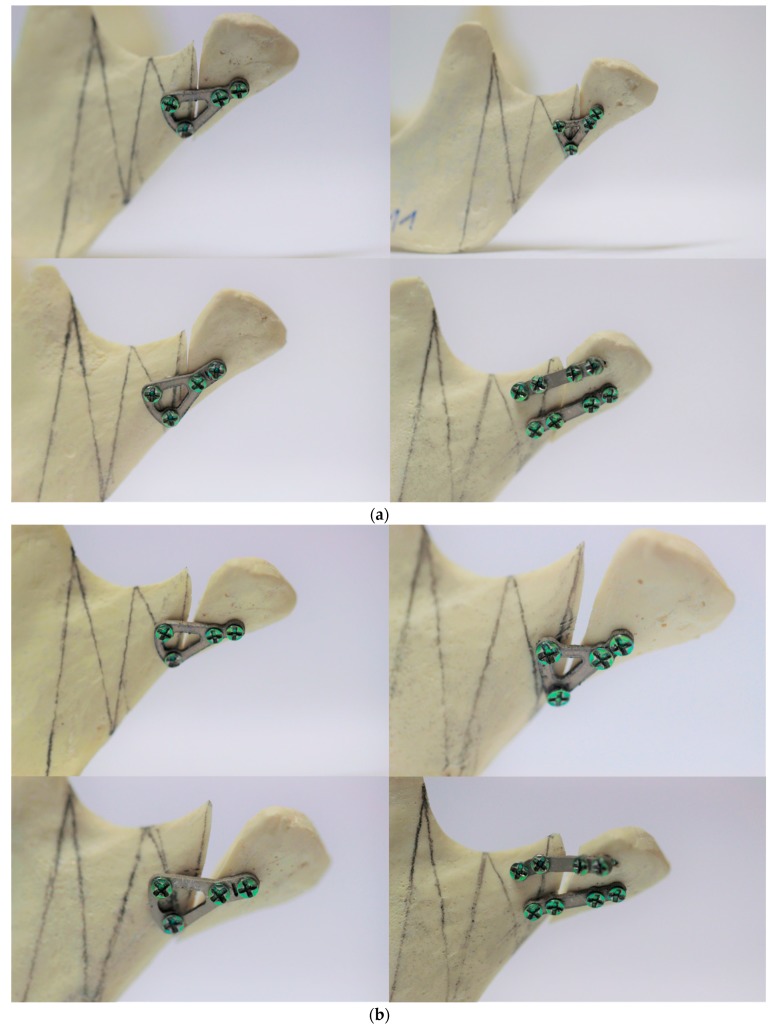
(**a**) Plates dedicated for high-neck fractures fixed to polyurethane mandibles before tests. (**b**) Plates dedicated for high-neck fractures fixed to polyurethane mandibles after tests. (**c**) Design 20 of plates during fatigued test.

**Figure 2 materials-13-00592-f002:**
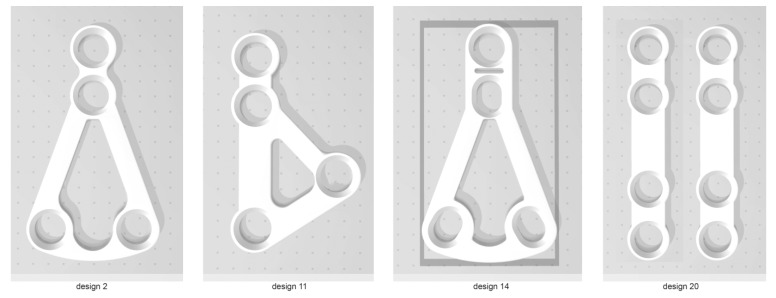
Design used in this study: screw system 2.0 mm. Design 20 is the golden standard of condylar osteosynthesis. Three other plates design feature by narrow upper part fitting anatomically to region of interest in mandible.

**Figure 3 materials-13-00592-f003:**
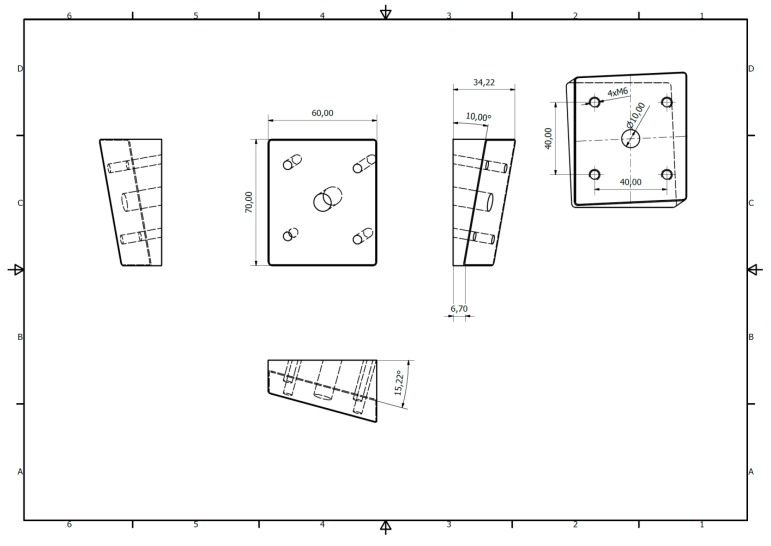
Stainless steel clamping system for fatigue tests of mandibles.

**Figure 4 materials-13-00592-f004:**
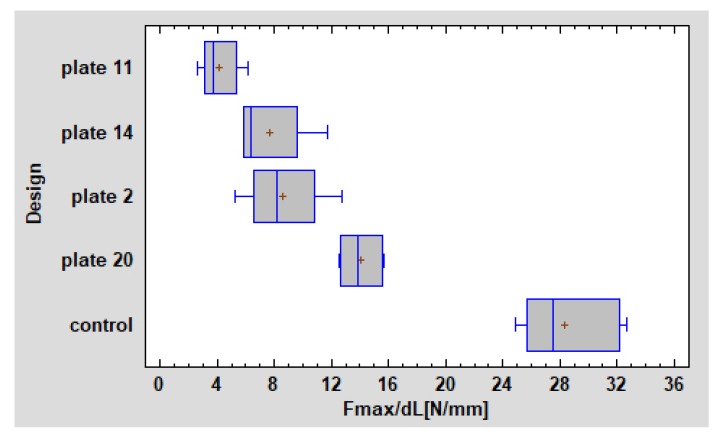
Fracture line as the maximal force used. Control - intact condyle. The weakest (plate 11) and the strongest (plates 20) osteosynthesis set.

**Figure 5 materials-13-00592-f005:**
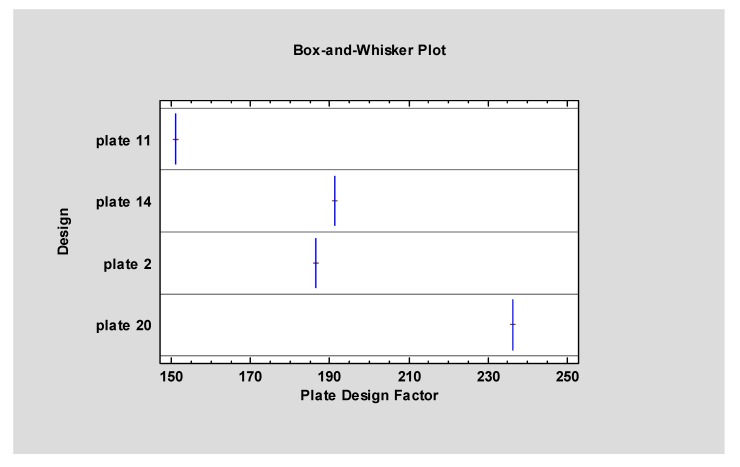
Plate Design Factor: plate 11 is 151, plate 14 is 191, plate 2 is 187 and plate 20 is 236. It proves plate 20 should be used as gold standard in high-neck fractures whereas plate 11 should fall into abeyance.

**Table 1 materials-13-00592-t001:** List of shapes of plates used in osteosynthesis of high-neck mandibular condylar fracture. Above 1 mm thick plates were cut by laser from medical certified titanium sheet alloy grade 23. Plate 11 (red cells) has been occurred as the weakest whereas plates 20 (green cells) have been the strongest.

Design Code	Manufacturer	Design	Height (mm)	Width (mm)
Plate11	Synthes	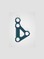	13,5	8
Plate14	Medartis	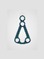	15,3	8,8
Plate02	Medartis	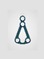	15,4	8,8
Plate20	any	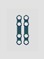	16,5	3,4

**Table 2 materials-13-00592-t002:** Analysis of fractures after solid screws fixation (Chi-Square test = 28, *p* < 0.0001). Only two single plates (plates 20) broke after rigid fixation. In others screws tend to break.

Plates No	Broken	Endured	Row Total
Plate 11	0	7	7
	0.00%	25.00%	25.00%
Plate 14	0	7	7
	0.00%	25.00%	25.00%
Plate 2	0	7	7
	0.00%	25.00%	25.00%
Plate 20	7	0	7
	25.00%	0.00%	25.00%
Column Total	7	21	28
	25.00%	75.00%	100.00%

**Table 3 materials-13-00592-t003:** Design data and summary statistics. Abbreviations: F max/dL – force required for one-millimeter displacement in fracture line after osteosynthesis.

One-millimeter Displacement in Fracture Line	Height (mm)	Width (mm)	Plate Surface Area (mm^2)^	Plate Design Factor	F max/dL (N/mm)
Average±stand. dev.	15.18±1.10	7.25±2.29	179.28±32.15	191.23±30.79	8.60±4.03
Minimum	13.5	3.4	138.13	151.05	2.57
Maximum	16.5	8.8	226.82	236.17	15.65
Range	3.0	5.4	88.69	85.13	13.08
Stnd. skewness	−1.08	−2.48	0.68	0.53	0.68
Stnd. kurtosis	−0.97	−0.68	−1.02	−1.03	−1.30

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
