# Peer review of "Mechanical Evaluation of Titanium Plates for Osteoesynthesis High Neck Condylar Fracture of Mandible"

_materials, 2020, doi:10.3390/ma13030592_

Round 1

Reviewer 1 Report

The manuscript entitled "Mechanical evaluation of titanium plates for osteoesynthesis high neck condylar fracture of mandible" by ZieliÅ„ski et al. reports mechanical tests on orthopaedic plates used for management of mandible fractures. The investigated plaques differ for shapes and the number of points of fixation (30 in total) on the mandibular body. For overcame issue regarding human mandible sampling for the test, authors use artificial mandible made with solid foam, a gold standard used for implants. Finally, they apply mechanical stress forces accordingly to guidelines reported in the literature. Only 13% of plaques result suitable for acceptable mechanical resilience. 

The paper focuses on a very specific field, apparently lacking a systematic comparison among the mandibular implants and it seems to be conducted properly despite the experimental limitations. Any concern arose on scientific soundness of the manuscript. Please correct some typo (i.e. osteoesynthesis, limmitation) or improper sentences.

Author Response

Dear Reviewer

I am very grateful for your answer and opinion. We will make suggested changes in the submission.

Yours faithfully

Rafal Zielinski

Reviewer 2 Report

Zieliński et al reported the mechanical evaluation of 4 plates for the high-neck mandibular condylar osteosynthesis using a new clamping system. The research is of practical significance, but the manuscript needs a significant improvement regarding the language. It is very difficult to read and understand. I recommend a major revision or resubmission.

My suggestions are

Table2. what does the percentage number mean? Table 2. Only two single plates [plates 20] means double plates or something else? Line 14, “all available designs (only 4 of 30 was proper) on polyurethane mandibles”. I would not say all available designs if you just evaluated 4 of them.

Language

Line 11 any to no

Line 20, “Double  straight  plates  the  best  mechanical  fixation  for  high-neck fractures  of  the  mandibular  condyle”.

Line 62 “In articles virtually no help a clinician may find.” is not readable

 Line 42 “destinied” is not readable

Line 189 what is FEM? 

Line 198 "Due to"  to because  

Line 199 desings to design

Line 200 develop to developed

Line 241 the double straight plates + are

Line 242, it to its

Author Response

Dear Reviewer

Thank you for your comments. 

In yellow my anwers:

My suggestions are

Table2. what does the percentage number mean? As description tells percentage mean "Analysis of fractures after solid screws fixation" so in percent we wrote how many of screws broke. Table 2. Only two single plates [plates 20] means double plates or something else? Yes, two single plates are gold standard in maxillofacial surgery in high-neck reconstruction that is why we wrote "only two single plates [plates 20]" Line 14, “all available designs (only 4 of 30 was proper) on polyurethane mandibles”. I would not say all available designs if you just evaluated 4 of them. Yes, you are right. But what can we do if only 4 designs are available on the market? It is true, we just evaluated in this research 4 types of designs but high-neck area is so specific that only 4 types are used by practitioners. Nevertheless, we can change "all available designs into "4 evaluated types of designs". In line 13 wrote "all available designs (only 4 of 30 was proper) " so it is clearly defined that "all available designs" means "4 types".

Language

Line 11 any to no - done

Line 20, “Double  straight  plates  the  best  mechanical  fixation  for  high-neck fractures  of  the  mandibular  condyle”. - I wrote "Double straight plates occurred to be the best mechanical fixation for high-neck fractures of the mandibular condyle"

Line 62 “In articles virtually no help a clinician may find.” is not readable - I changed it into "In the literature it is impossible to find out which type of designs is proper for high-neck fratures."

 Line 42 “destinied” is not readable - I changed it into "However, no studies could be found with comparison of osteosynthesis plates dedicated to condylar neck fractures."

Line 189 what is FEM? - I changed "FEM - Analysis" into "Finite Element Method (FEM)"

Line 198 "Due to"  to because  - done

Line 199 desings to design - done 

Line 200 develop to developed - done 

Line 241 the double straight plates + are - done

Line 242, it to its - done

Thank you for your comments. I hope you accept the manuscript and we will be able to go further and English supervisor indicate more English mistakes that I could change them. I have already published 2 manuscripts in Materials and always after Reviewers opinion English supervisor asks for corrections.

Yours faithfully

Rafal Zielinski

Reviewer 3 Report

The work is well written and minor revision is suggested.

Figure 4 and 5 can be re-depicted by a better software.

Author Response

Dear Reviewer

Thank you for your opinion. We will make suggested changes.

Regards

Rafal Zielinski

Round 2

Reviewer 2 Report

Most of my concerns have been addressed. Recommend to publish. 

This manuscript is a resubmission of an earlier submission. The following is a list of the peer review reports and author responses from that submission.